# *Escherichia coli* O157:H7 F9 Fimbriae Recognize Plant Xyloglucan and Elicit a Response in *Arabidopsis thaliana*

**DOI:** 10.3390/ijms21249720

**Published:** 2020-12-19

**Authors:** Ashleigh Holmes, Yannick Rossez, Kathryn Mary Wright, Pete Edward Hedley, Jenny Morris, William George Tycho Willats, Nicola Jean Holden

**Affiliations:** 1Cell & Molecular Sciences, The James Hutton Institute, Dundee DD2 5DA, UK; Ashleigh.Holmes@hutton.ac.uk (A.H.); yannick.rossez@utc.fr (Y.R.); Kath.Wright@hutton.ac.uk (K.M.W.); Pete.Hedley@hutton.ac.uk (P.E.H.); Jenny.Morris@hutton.ac.uk (J.M.); 2Enzyme and Cell Engineering UMR 7025 CNRS, Université de Technologie de Compiègne, CEDEX, 60205 Compiègne, France; 3School of Natural and Environmental Sciences, Newcastle University, Newcastle upon Tyne NE1 7RU, UK; William.Willats@newcastle.ac.uk; 4SRUC, Department of Rural Land Use, Craibstone Estate, Aberdeen AB21 9YA, UK

**Keywords:** *Escherichia coli*, host–microbe interaction, bacterial adhesion, fimbriae, glycan array, xyloglucan, plant defence, ELISA, immunofluorescence microscopy

## Abstract

Fresh produce is often a source of enterohaemorrhagic *Escherichia coli* (EHEC) outbreaks. Fimbriae are extracellular structures involved in cell-to-cell attachment and surface colonisation. F9 (Fml) fimbriae have been shown to be expressed at temperatures lower than 37 °C, implying a function beyond the mammalian host. We demonstrate that F9 fimbriae recognize plant cell wall hemicellulose, specifically galactosylated side chains of xyloglucan, using glycan arrays. *E. coli* expressing F9 fimbriae had a positive advantage for adherence to spinach hemicellulose extract and tissues, which have galactosylated oligosaccharides as recognized by LM24 and LM25 antibodies. As fimbriae are multimeric structures with a molecular pattern, we investigated whether F9 fimbriae could induce a transcriptional response in model plant *Arabidopsis thaliana*, compared with flagella and another fimbrial type, *E. coli* common pilus (ECP), using DNA microarrays. F9 induced the differential expression of 435 genes, including genes involved in the plant defence response. The expression of F9 at environmentally relevant temperatures and its recognition of plant xyloglucan adds to the suite of adhesins EHEC has available to exploit the plant niche.

## 1. Introduction

Enterohaemorrhagic *Escherichia coli* (EHEC) serotype O157:H7 are responsible for food-borne diseases including haemorrhagic colitis or life-threatening complications, such as haemolytic uremic syndrome (HUS) [1], and are frequently associated with outbreaks linked to fresh produce [2]. A large-scale outbreak in Japan was directly linked with consumption of radish sprouts contaminated by a strain of EHEC O157:H7 Sakai [3].

To efficiently colonise tissues or surfaces, bacteria need to adhere via proteinaceous structures on the cell surface, termed adhesins [4]. Type 1 and P fimbriae are the best characterized structures and belong to a large family of bacterial adhesins, defined by their secretion mechanism, the chaperone-usher pathway (CUP). The CUP is well conserved in Gram-negative bacteria. The receptor-binding adhesin occupies the distal end of the fimbrial organelle and connects the adhesin to the terminal major subunit protein. Fimbrial adhesins mediate binding to specific ligands over a lectin domain [5]. Type 1 fimbriae adhesin (FimH) binds to mannosylated receptors (α1-3 mannan) via *N*-linked glycans [6]. FimH can be expressed at low temperatures, compatible with plant growth, and has been shown to target mannose containing polysaccharides and N-glycans from plants [7]. Similarly, Yad fimbriae from *E. coli* K-12 and *E. coli* common pilus (ECP) are expressed at a low temperature and mediate binding to plant cell walls via xylose and arabinans, respectively [8,9]. Phylogenetic analysis of *E. coli* identified another CUP gene cluster, F9 fimbriae (also known as Fml/Yde), as part of the γ1 fimbrial subclade, which formed a monophyletic cluster with type 1 fimbriae, suggesting that they originated through gene duplication from a common ancestral operon [10].

The F9 gene cluster is not ubiquitous in *E. coli* and is restricted to three main groups: (i) present in isolates in the same gene cluster organisation as EHEC Sakai; (ii) present in the environmental isolate *E. coli* SMS-3-5, although this version does not share the same putative adhesin; and (iii) present in ExPEC and described for UPEC (isolate CFT073), with a different regulatory organisation compared with the EHEC cluster-type that contains an apparent insertion and different transcriptional regulator [11]. Unlike some other adhesin gene clusters (e.g., Pap), the EHEC F9 cluster is not associated with insertion into *t*RNA sites (as determined by Island Viewer 4 [12]), and is thus likely to have arisen through recombination. F9 fimbriae was first described for EHEC to play a role in adhesion on bovine epithelial cells [13] and later in biofilm formation for Uropathogenic *Escherichia coli* (UPEC) [14]. F9 expression has been shown to occur at low temperatures (around 20 °C) [11,15]. F9 fimbriae mediate adhesion to terminal d-galactose linked in *β*1-3 with a residue of *N*-acetyl d-glucosamine (Gal*β*1-3GlcNAc) sequence, also defined as type 1 *N-*acetyllactosamine (LacNAc type 1) [11]. Terminal LacNAc type 1 has been found in core-1 and -2 *O*-glycans of inflamed urothelium, associated with UPEC persistence during chronic cystitis in an acute mouse model of cystitis. Furthermore, vaccination with FmlH was shown to protect mice from chronic cystitis [16].

In contrast to animal cells, plant cells are surrounded by a polysaccharide-rich wall. These walls consist mainly of cellulose, non-cellulosic polysaccharides including pectins, hemicelluloses, and proteoglycans, and in the case of secondary walls, lignin. Hemicelluloses are structural components of walls, usually branched [17] and divided into four general groups depending on backbone composition; that is, xylans, mannans, mixed linkage β-glucans, and xyloglucans [18]. Xyloglucans are the most abundant hemicelluloses of primary cell walls from higher plant species [19]. The structure-based nomenclature for xyloglucan-derived oligosaccharides describes the attachments to the backbone: unbranched glucosyl residue is denoted G; a glucosyl residue linked to a single xylose is denoted X; and, when linked to a disaccharide of *β*Gal1, 2*α*Xyl is denoted L, e.g., XXXG stands for three consecutive glucosyl residues with xylose attached and a fourth unbranched residue [20].

Plants perceive microbes via microbe-associated molecular patterns (MAMPs), sometimes also termed pathogen-associated molecular patterns (PAMPs). MAMP recognition is the basal defence and can lead to more specific responses involving plant R (resistance) proteins, which recognize intracellular pathogen effectors (reviewed in [21]). Bacterial flagella are almost ubiquitous and the flagellin monomer is a potent (and well-described) MAMP [22,23]. Other bacterial MAMPs include elongation factor Tu (EF-Tu) [24], peptidoglycan [25], cold shock proteins [26], and lipopolysaccharide [27] (all reviewed in [28]). As yet, fimbriae have not been described to elicit an immune response in plants, but have been heterologously expressed in plants for animal vaccine candidates [29,30].

Here, we demonstrate that F9 from EHEC O157:H7 strain Sakai can bind to spinach tissues through hemicellulose polysaccharide harbouring galactose. We also show that F9 fimbriae invoke a host response in *Arabidopsis thaliana* that is distinct from the response to either H7 flagella or ECP fimbriae.

## 2. Results

### 2.1. F9 Description

The *f9* gene cluster of EHEC Sakai (Appendix A; Genbank # BA000007.3) does not include a canonical in cis transcriptional regulator, as the open reading frame (ORF) upstream of the main structural subunit, *fmlA* (ECs2113), is an incomplete allele of *hipA* without its cognate partner *hipB*, a toxin/antitoxin system [31,32]. However, a cryptic CDS (ECs2114) is located between *hipA* and *fmlA* that belongs to the AraC-like regulator family, containing the HTH_18 superfamily domain, and is present as a hypothetical gene in the equivalent EDL933 genome (z2199). The Sakai and EDL933 amino acid sequences of the Z2199 regulators are 100% conserved, as are the 5′-untranslated regions (UTRs), but with two potential start codons.

### 2.2. F9 (Regulator and FimA-Like) GFP+ Transcriptional Fusion Expression In Vitro

F9 reporter plasmid-transformed EHEC Sakai were grown under conditions to indicate potential catabolite control (i.e., glucose ‘vs.’ glycerol) and at a plant-relevant temperature of 18 °C, compared with 37 °C (Figure 1). The putative regulator *ECs2114* reporter showed minimal expression in both media types and temperatures, although there was some evidence for higher expression at 37 °C (230 ± 144) compared with 18 °C (101 ± 64). The translational fusion of *fmlA* was induced compared with the transcriptional fusion, for all conditions except glucose, at 37 °C. There appears to be temperature-dependent translational control of *fmlA*, as expression was at least sixfold higher at 18 °C (glucose = 15,777 ± 1628; glycerol = 8322 ± 942 average ± SD).

### 2.3. Functional Analysis of EHEC F9 Fimbriae

#### 2.3.1. Recognition of Plant Glycans

In order to identify any potential plant–ligand binding targets for F9 fimbriae, a high-throughput screen using plant glycan arrays (polysaccharide and oligosaccharides) was used, as previously for ECP fimbriae [8]. F9 organelles were overexpressed and purified from *E. coli* strain JT-1 (Appendix A). Purified F9 fimbriae interacted with galactosylated xyloglucans, but not fucosylated xyloglucans (FLX) (Figure 2A, blue box). They also bound to oligosaccharides XLX and LLX, but not with XXX (Figure 2A, red box). The binding overlapped with glycan specific antibody LM15 (anti-xyloglucan, Figure 2B), confirming xyloglucan epitopes on the array.

#### 2.3.2. Interaction with Spinach Hemicellulose Extract and Tissues

Xyloglucan motifs were differentially distributed in the leaves compared with roots for spinach, a horticultural plant relevant to EHEC foodborne outbreaks, such that there were greater levels of XXXG in root (LM15 and LM25) compared with leaf hemicellulose-enriched extracts (Figure 3A). In contrast, there was more XLLG (LM24) in spinach leaf compared with root extracts. *E. coli* (JT-1) overexpressing F9 (pASL04) adhered significantly better to leaf hemicellulose extract compared with the vector only control (pBAD18). Overexpressing F9 did not provide a significant advantage for adherence to root extracts, but did show a positive trend (Figure 3B). A defined mutant of the main F9 structural subunit in EHEC Sakai (*fmlA/ECs2113*) was constructed to test a functional role for adherence (Figure 3C). EHEC Sakai lacking the F9 fimbriae were recovered from spinach leaf and root tissues at a significantly lower rate compared with the isogenic parent (one-way analysis of variance (ANOVA); Dunnett’s multiple comparison test), although the decrease was small. Adherence was restored to WT levels in the F9 mutant complemented with a low copy plasmid containing *fmlA* sequence (pAH012). The average number of F9 mutant with the empty control vector recovered from spinach leaves (Figure 3Ci) was significantly different from the WT, but not the complemented strain (pAH012) (one-way ANOVA; Dunnett’s multiple comparison test). Thus, there was an increase in the average number of bacteria recovered from spinach leaves in the presence of FmlA (ECs2113). The positive effect was not significant on spinach roots because transformation of the mutant strain with either plasmid increased the number of bacteria recovered (Figure 3Cii). This may be down to artefactual effects from the plasmid transformation; expression variation of *fmlA* from the low-copy vector, or a combination of both. Bacteria resistant to ampicillin and/or tetracycline were not recovered from uninfected control plant tissues.

#### 2.3.3. Characterisation of Spinach Leaf Xyloglucan in Situ

To understand the distribution of galactosylated xyloglucan in spinach leaves, the distribution of LM15, LM24, and LM25 epitopes was determined by immunofluorescence microscopy (Figure 4). The preliminary results demonstrated labelling only with LM25, and thus tissue was subsequently pre-treated with pectate lyase in case pectin in the cell wall masked the hemicellulose epitopes. LM25 shows the greatest labelling of the spinach leaf abaxial epidermal surface, localizing over the cell walls and at the guard cells of the stomata (Figure 4A). Following pectate lyase treatment, LM24 addition resulted in large aggregates of labelling (Figure 4E), while LM15 resulted in more diffuse labelling, particularly aligned with the anticlinal walls (Figure 4F). The addition of enzymatic/high pH treatment reduced LM25 labelling of the epidermal cell walls, but not the guard cells (Figure 4D). Fluorescent punctae were observed in untreated LM24 (Figure 4B) and LM15 (Figure 4C) tissue and control tissue incubated without primary antibody (Figure 4G–I).

### 2.4. Plants Can Elicit a Response to F9 Fimbriae

We wanted to investigate whether purified F9 fimbriae elicited a MAMP response in plants. We chose a microarray transcriptomics approach using *Arabidopsis thaliana* as a well-established model for plant defence responses, particularly MAMP triggered immunity (MTI). The response to F9 was compared to purified H7 flagella; the flagellin monomer is a well-characterized MAMP, but here, we used the multimeric form rather than the classic eliciting 22 amino acid epitope, Flg22 [22,23]. We have previously described the role of *E. coli* common pilus (ECP) in the attachment of E. coli O157:H7 to plant tissues, specifically to arabinan containing residues [8], and thus included it for a comparison of the plant response to a more common fimbrial type.

#### 2.4.1. Global Transcriptomic Response in *Arabidopsis thaliana*

Infiltration of purified F9 resulted in the significant (*p* > 0.05) differential expression of 435 *A. thaliana* genes compared with 1521 in response to H7 flagella or 320 genes with ECP fimbriae (Figure 5, Appendix A). Treatment with F9 resulted in upregulation of 176 genes and 259 repressed genes. Of the F9-responsive genes, 48% of genes were specifically differentially expressed compared with 78% of H7-responsive genes or 37% of ECP-responsive genes. There is little commonality across all three conditions, particularly for repressed genes. Only 1.2% (23) of differentially expressed, H7-independent genes were shared by F9 and ECP; these genes may be described as fimbrial-responsive.

#### 2.4.2. Comparisons with Other Published Datasets

In order to define the genes associated with microbe/pathogen-associated molecular pattern trigger immunity (MTI/PTI), we compared our dataset to previous publications that had investigated the MTI response to *E. coli* and *Salmonella enterica*. Schikora et al. [34] investigated the transcriptomic response of *A. thaliana* seedlings to the non-pathogenic isolate of *E. coli* DH5α (Appendix A). Of the 289 genes described as responsive to *E. coli* K-12 treatment, 23 genes were also responsive to F9, 79 genes to H7, and 38 genes to ECP (Figure 6A). *E. coli* isolate DH5α encode flagella serotype H48 rather than EHEC (Sakai) H7 serotype, although the Flg22 epitope is conserved in both [22,23]. Comparison with an *A. thaliana* dataset that focused on the presence of flagella and included *E. coli* O157:H7 *fliC*- [35] identified 49 F9-responsive genes out of the 748 genes described, 249 with H7, and 89 genes with ECP, of which 17 were in common (Figure 6B, Appendix A). Our dataset showed that the overall change in *A. thaliana* gene expression was the same, i.e., 92–98%, either positively or negatively impacted. Of the total number of genes differentially regulated for each treatment, MAMP response genes constitute 11% of the response to F9 fimbriae, 16% to H7 flagella, and 28% to ECP fimbriae.

To examine the F9-specific response in more detail, MapMan [36] was used to assess the genes involved in biotic stress response (Figure 6C). Out of 435 genes, 112 mapped, including those involved in jasmonic acid signaling, PR defence proteins, peroxidases, and cell wall turnover. The H7-specific response elicited four times more (462) genes for the biotic response, whereas the ECP-specific response was more comparable (96) (Appendix A). Infiltration of *Arabidopsis* leaves with purified H7 invoked the significant upregulated expression of well characterized flagellin response genes FRK1 (AT2G19190), FLS2 (AT5G46330), WRKY22 (AT4G01250), WRKY29 (AT4G23550), and PAL1 (AT2G37040) [37,38,39]. ECP also stimulated upregulation of FRK1.

## 3. Discussion

F9 fimbriae are not common across *Escherichia coli,* but limited to pathogenic EHEC and UPEC isolates. Environmental *E. coli* isolate SMS encodes an orthologous cluster that has a divergent adhesin compared with the pathogenic counterpart. Previous work has shown that F9 fimbriae from EHEC can be expressed at low temperatures [13] and it was identified in a genetic screen for adherence to calf intestines [40]. In EHEC Sakai, there is a putative regulator (*ECs2114*, Appendix A) with predicted HTH DNA binding domain encoded upstream of the fimbrial operon, which was not induced under the conditions tested within this study. Here, the translational reporter fusion showed temperature-dependent control of F9 main subunit (*fmlA*) expression, which supports previous reports. The induction of *fmlA* expression in glycerol at 37 °C from the translational reporter compared with the transcriptional reporter may be because of the different GFP variant used in the constructs.

Interestingly, F9 specifically recognize non-fucosylated xyloglucans (representative from tamarind seeds and not from peas). Although F9 binding overlapped with the plant probe LM15 on the glycan array, LM15 does not distinguish between these two types of xyloglucans, suggesting a different mechanism of binding (Figure 2). However, it is possible that fucose masks the epitope harbored by the fucosylated xyloglucan. Interaction with xyloglucan oligosaccharides provides more insight about the binding recognition of F9. While LM15 recognized LLX, XLX, and XXX, F9 binds only to LLX and XLX. These two oligosaccharides bear two or one (respectively) *β*-galactose linked in 1–2 to a residue of xylose in the α position (Gal1,2*α*Xyl). In mammalian tissues, F9 can bind specifically to terminal d-galactose, but is linked in *β*1–3 to *N*-acetyl d-glucosamine (Gal*β*1-3GlcNAc) [10,16]. These carbohydrates from animals and plants have a terminal galactose in common, but with different linkages. This result is in accordance with the lectin specificity of the adhesin of F9 previously described.

Spinach leaf polysaccharide extracts contained significantly more xyloglucan oligosaccharides recognized by LM24 than by LM25 or LM15 (Figure 3A). It is noteworthy that LM24 is known to bind more efficiently to LLX oligosaccharide than XXX and XLX, and at a higher level compared with LM15 and LM25 [41]. This makes LM24 the more specific antibody, and to our knowledge, for galactosylated xyloglucan oligomers. *E. coli* expressing F9 conferred a positive advantage for adherence to spinach leaf extract, although the EHEC Sakai F9 mutant was negatively impacted for adherence to both spinach leaf and root tissues. This may be down to differences in F9 binding targets, expression of other adherence factors, or a combination of both. Immunofluorescence labelling of spinach leaf tissue with LM25 shows that there are galactosyl residues (LXX/LLX) presented for the interaction of F9 on the epidermal surface and guard cells (Figure 4). The greater specificity of LM24 for LLX oligomers may explain the requirement for exposure by pectic lyase treatment.

F9 not only recognizes specific hemicellulose ligands, but it is also perceived by the plant host in a specific manner. *Arabidopsis thaliana* is a well-established plant model and studies have shown that non-fucosylated xyloglucan is present in the primary cell wall of *Arabidopsis* mesophyll protoplasts [42] and galactosylated xyloglucan oligosaccharides constitute approximately 25% of total stem xyloglucan [43]. LM24 was reported to strongly label the outer layer of *Arabidopsis* apical shoot meristems [44] and demonstrates the presence of LLX oligosaccharides in *Arabidopsis* leaf hemicellulose extract [45]. The global transcriptomic response of *A. thaliana* to purified *E. coli* organelles was greatest for H7 flagella (1521 genes), as anticipated for a major MAMP, compared with F9 (435 genes) and ECP (320 genes) fimbriae (Figure 5). Approximately half of the F9-responsive genes are shared with the other treatments, indicating similar pathways may be affected by bacterial organelles either as part of a general response to *E. coli* or a MTI response.

Only 13 genes are common between the three treatments and the general *E. coli* response subset (Figure 6A), indicating that the response of the plant to whole *E. coli* cells is multifactorial, and that different subsets of genes may be affected by the different epitopes present. The vast majority of *A. thaliana* MTI related genes (as defined by published datasets [34,35]) induced by F9 were also induced by other EHEC surface organelles, H7 and ECP, indicative of a common MTI response (Figure 6B). These included genes induced by Flg22, such as pectin methylesterase 17 (PME17; AT2G45220) [46] and FAD-linked oxidoreductase gene CELLOX (AT4G20860) [47]. Interestingly, one FAD-linked gene, BBE8 (AT1G30700), induced by all three *E. coli* organelles, is involved in stomatal opening by *S. enterica* and *Pseudomonas syringae* [48].

F9 induced a specific response for putative plant defensin protein, (PDF1.4, AT1G19610), which is predicted to encode a PR (pathogenesis-related) protein. Plant defensins have a well-characterized role in antimicrobial responses against a range of bacteria and fungi [49]. PDF1.4 has been shown to be induced in the non-host response of *A. thaliana* to barley powdery mildew [50], thus it is intriguing that this defensin was only triggered by F9 and not H7 or ECP. Instead, H7 and ECP induced another member of the defensin family, PDF1.1. PDF1.4 is cytoplasmic, whereas PDF1.1 is secreted and not induced by methyl-jasmonate [50]. F9 downregulated PDCB3 (AT1G18650), a member of GPI-anchored protein family that localises to the apoplastic face of plasmodesmata and bind callose, affecting cell-to-cell traffic [51]. Although PDCB3 expression, as measured by q-RT PCR, was not impacted by heat or wound treatment [51], it may be influenced by lectin interactions with the plant cell wall. Furthermore, both F9 and ECP induced SHB1 (AT4G25350), which acts in cellular signaling and is associated with cell elongation [52], suggesting that fimbriae that recognize cell wall ligands trigger a separate signaling pathway. There is mounting evidence that the plant immune response is not only dependent upon the plasma membrane bound receptors, but also on plant cell wall integrity [53]. It would be interesting to investigate whether the plant responses reported here are due to an interruption of cell wall integrity by fimbrial interactions with the plant cell wall components.

Although F9 binds to spinach tissue, it is most likely to act as a part of an adherence consortium, and on its own does not have a marked phenotype. Nonetheless, it is the combination of the adherence complement plus available ligands that results in the outcome of specific colonization and localization on host tissue. Here, we describe the interaction of F9 with hemicellulose component xyloglucan and the results herein could offer a reason to why EHEC foodborne outbreaks are often associated with spinach [54,55]. Previous work has also identified plant carbohydrate ligands for ubiquitous *E. coli* fimbriae ECP [8], Type 1 fimbriae [6], and Yad fimbriae [9], which are differentially expressed. Thus, the specific complement of adherence factors encoded by EHEC isolates sets them apart from other *E. coli*, and thereby may contribute to niche exclusion.

## 4. Materials and Methods 

### 4.1. Bacterial Strains and Growth Conditions

*Escherichia coli* O157:H7 strain Sakai (RIMD 0509952; [56]) stx^−^ Kan^R^, herein termed EHEC Sakai, and *E. coli* K-12 strain JT-1 [57] were grown with aeration at 200 rpm in either lysogeny broth (LB) or MOPS medium [58] supplemented with 0.2% glucose (or glycerol where indicated), 10 μM thiamine, and MEM essential and non-essential amino acids (Sigma M5550 and M7145) termed rich defined MOPS (RD-MOPS) media. Antibiotics were included where necessary at the following concentrations: 50 μg/mL kanamycin (Kan), 12.5 μg/mL chloramphenicol (Cam), 10 μg/mL Tetracycline (Tet), and 50 μg/mL ampicillin (Amp).

### 4.2. Molecular Methods

Plasmids and primers are listed in Table 1. For the GFP transcriptional reporters, the 5′UTR of *fmlA* (441 nt; ECs2113) and *ECs2114* was PCR amplified (primers 2113utr_for, 2113utr_rev, 2114utr_for, 2114utr_rev) with Phusion polymerase from New England Biolabs (Hitchin, UK) and cloned into pKC026 using XbaI, creating the transcriptional fusions pAH010 and pAH011, respectively. There are two potential start codons for putative regulator *ECs2114*, so a transcriptional fusion was generated from the longer version of the regulator, encompassing 274 nt of the 5′-UTR. A defined deletion in the EHEC Sakai *fmlA* gene (ECs2113) was constructed using allelic exchange as previously described [8,59] using constructed vector pAH013. Deletions were confirmed by PCR and Sanger sequencing. To complement the mutation in trans, *fmlA* and 500 bp of its flanking sequence was PCR amplified (2113.5F and 2113.3R) and cloned into low copy plasmid pWSK29, via pGEMT-easy, to create pAH012. Restriction enzymes and T4 DNA ligase used were from Promega (Southampton, UK).

### 4.3. Analysis of Bacterial Fluorescence In Vitro

Gene expression was measured from EHEC Sakai transformed with pAH010, pAH011, or pACloc8 following growth for ~18 h in LB medium + Cam at 37 °C, 200 rpm before diluting 1:100 into 15 mL RD MOPS medium supplemented with 0.2% glucose or glycerol. Cultures were incubated shaking at 18 °C or 37 °C and measured for cell density and GFP fluorescence after 24 h. GFP fluorescence was measured in triplicate 150 µL volumes in a 96-well plate using GloMax plate reader (Promega, Southampton, UK). EHEC Sakai transformed with the vector control plasmid pKC026 or pAJR70 (for pACloc8) was included as a control for background fluorescence. Fluorescence was normalised to cell density (OD_600_) and background subtracted, and values were plotted using GraphPad Prism version 5.00 for Windows (GraphPad Software, San Diego, CA, USA) for three experimental repeats.

### 4.4. Protein Purification and Antibody Production

H7 serotype flagella were purified as described previously from pEBW3 transformed into JT-1 [64]. ECP fimbriae were purified from pMAT3 and F9 fimbriae from pASL04 transformed into JT-1 as described previously [8]. All protein preparations were detoxified using Pierce High-Capacity Endotoxin removal spin column (ThermoScientific, Perth, UK) as per the manufacturer’s instructions.

Rabbit polyclonal antibodies were produced against a peptide (DSESMNQTVELGQVRSSRLC) from F9 major subunit according to standard procedures by Genoshere Biotechnologies (France).

Protein concentrations were determined using the BCA protein assay (Thermo Fisher Scientific Inc., Waltham, MA, USA).

### 4.5. Glycan Array

Glycan array printing and screening were performed as described previously [41,64]. All oligosaccharides are listed in Table 2 and glycan array layout shown in Figure 7. The results are based on three individual experiments.

### 4.6. Plant Growth Conditions and Extract Preparation

Spinach (*Spinacia oleracea*, var. Amazon) was grown in individual pots in a glasshouse at 22 °C (16 h light, 8 h dark), with 130–150 µmol m^−2^ s^−1^ light intensity. Compost was used for all experiments except for plant root polysaccharides extraction, where vermiculite was supplemented with Osmocote Start NPK (6 weeks), containing 12-11-17, to avoid compost contamination (peat).

*Arabidopsis thaliana* Col-0 were surfaced sterilised with Triton X-100 and ethanol and sown on Murashige–Skoog + 20% (*w*/*v*) sucrose agar plates. Seeds were vernalised at 4 °C for 2–3 days in the dark and then transferred to a growth cabinet (22 °C; 16 h light, 8 h dark photoperiod) for a further 11–12 days. Seedlings were transplanted four plants per 8 cm round pot, for another 3 weeks’ growth in the cabinet. Four to five-week old plants were used for infiltration experiments.

Plant polysaccharide extracts were prepared as described in [8]. In short, alcohol insoluble residue was prepared by five washes of homogenised spinach leaves or roots with 70% ethanol 6:1 (*v*/*v*), followed by two acetone washes, and then the pellet was air-dried. Sequential extraction of pectic and hemicellulosic fractions was performed by incubating the insoluble fraction in 50 mM diamino-cyclo-hexane-tetra-acetic acid (CDTA), pH 7.5, and 4 M NaOH, respectively. Polysaccharide content was quantified by a phenol–sulfuric acid method in microplate format [64] after acid hydrolysis (2 M trifluoroacetic acid for 1 h at 120 °C).

### 4.7. Enzyme Linked Immuno-Sorbent Assay (ELISA)

ELISA was performed as described in [8]. NUNC Maxisorp 96-well plates were coated with 5 μg polysaccharides (in Tris buffered saline (TBS)) overnight at 4 °C and then incubated with ~1 × 10^6^ cfu bacteria or 1/20 dilution monoclonal antibody. Primary monoclonal antibodies used in this study include LM15 [19], LM24, and LM25 [41] (Plant Probes, Leeds, UK). Bacteria were detected with 1/500 dilution rabbit anti-*E. coli* antibody (Abcam, Cambridge, UK). Secondary HRP-linked antibodies were used at 1/1000 dilution. Colour reaction was developed with 2,2′-azino-di-(3-ethylbenzthiazoline sulfonic acid (ABTS) solution: 22 mg ABTS (Sigma Aldrich, St. Louis, MO, USA) diluted in 100 mL of citrate buffer (50 mM sodium citrate, 0.05% H_2_O_2_, pH 4.0). The absorbance at OD_405_ was measured on a microplate reader SpectraMax M5 (Molecular Devices, Sunnyvale, CA, USA).

### 4.8. Bacterial Adhesion Assays

Adherence assays were performed as described in [65]. In short, pre-weighed spinach plant tissues were incubated in bacterial suspension (~1 × 10^7^ cfu/mL in sterile PBS; OD_600_ = 0.02) for two hours at 18 °C, 80 rpm. Plant samples were vigorously washed three times in sterile PBS by mixing on a vortexer, and then homogenised with a sterile pestle and mortar. Samples were serially diluted and plated on MacConkey’s agar with appropriate antibiotics for viable counts. Each experiment used five plants per bacterial strain and the experiment was repeated three times.

### 4.9. Immunofluorescence Microscopy

Leaf discs, 7.6 mm in diameter, were cut from young spinach leaves 4–6 cm in length and the abaxial surface floated on sterile distilled water (control). Samples were pre-treated either in CAPS buffer (50 mM CAPS, 2 mM CaCl_2_, adjusted to pH 10 with KOH) or buffer plus 10 µg/mL Pectate lyase (from *C. japonicus*; Megazyme, Bray, Ireland) at 37 °C for 2 h [66]. Control and pre-treated discs were labelled with antibody using methods modified from [67], by blocking in 2% BSA in microtubule stabilizing buffer (MTSB) for 2 h at 37 °C before addition of antibody, LM15, LM24, or LM25, diluted 1:50 in BSA+ MTSB and incubation overnight at 6 °C. The discs were washed three times in MTSB before addition of secondary antibody Alexa Fluor^®^ 488 goat anti rat (ThermoFisher Scientific, Perth, UK) diluted 1:500 in BSA+ MTSB and incubation for 2 h at 37 °C. After washing three times for 5 min in MTSB, discs were mounted abaxial surface up in MTSB, and observed using a Nikon A1R confocal laser scanning microscope with an NIR Apo 40X DIC water dipping lens at 488 nm excitation and emission from Alexa Fluor^®^ 488 collected between 500–530 nm and chlorophyll 663–737 nm with the gain set to prevent saturation (blanks collected at the same gains as labelled tissue). Images were assembled, equally linearly enhanced, and reduced in size using Adobe Photoshop Elements 2019 (17.0 X64).

### 4.10. Treatment of Plants with Protein Preparations

Fimbriae (1 μM) and H7 flagella (750 nM) were pressure infiltrated by needleless syringe into the leaf tissue. Then, 10 mM Tris-HCl + 0.2 M NaCl (endotoxin free) buffer, used for eluting from the detoxification columns, was used as a control. After 3 h of incubation, the infiltrated leaf was dropped into liquid nitrogen, ground to a powder by mortar and pestle, and transferred to a pre-chilled Eppendorf. Samples were kept at −80 °C until RNA purification. For each treatment, three individual plant replicates were used.

### 4.11. RNA Purification and Microarray

RNA was purified from samples using Qiagen RNAeasy Plant Mini kit following the manufacturer’s instructions and including the DNase treatment step (Qiagen, Manchester, UK). RNA concentration and quality were checked on BioAnalyzer2100 (Agilent Technologies, Cheadle, UK). Single-colour labelling, hybridization, and washing of the Agilent Arabidopsis (v. 4) Gene Expression 4x44k microarray slides were performed according to the manufacturer’s protocols (One-Color Microarray-Based Gene Expression Analysis, v. 6.5; Agilent Technologies). Microarrays were scanned using a G2505B scanner (Agilent Technologies, Cheadle, UK) as recommended and data were extracted from images using Feature Extraction software (Agilent v. 10.7.3.1) with default parameters. Data are available at ArrayExpress (https://www.ebi.ac.uk/arrayexpress/; E-MTAB-9732). Data were subsequently imported into GeneSpring GX 7.3 (Agilent Technologies, Santa Clara, CA, USA). Quality control was applied to remove those probes with no consistent signal in any of the conditions tested, whereby data were filtered on flags present or marginal in at least two out of the twelve arrays. Principal component analysis was performed to identify any outlier samples. Statistical analysis of the datasets was carried out by performing a Volcano plot on each condition with a twofold minimum cut-off for fold change and a Student’s *t*-test with multiple testing correction (Benjamini and Hochberg; *p* < 0.05). A heatmap of the significantly changing datasets was generated using the Gene Tree function in GeneSpring, and default parameters for clustering (Pearson’s correlation).

Microarray data were validated by qPCR as described in [68] for the following subset of significantly regulated genes: MAPKKK15 (AT5G55090), AT1G10360, AT4G12500, AT1G22900, AT2G17740, WRKY29 (AT4G23550), AT3G61930, AT5G44910, and AT1G33760. Correlation coefficients were 0.9635, 0.9581, and 0.9302 for F9, ECP, and H7, respectively.

Venn diagrams were constructed using Venny 2.1.0 online tool [69].

## Figures and Tables

**Figure 1 ijms-21-09720-f001:**
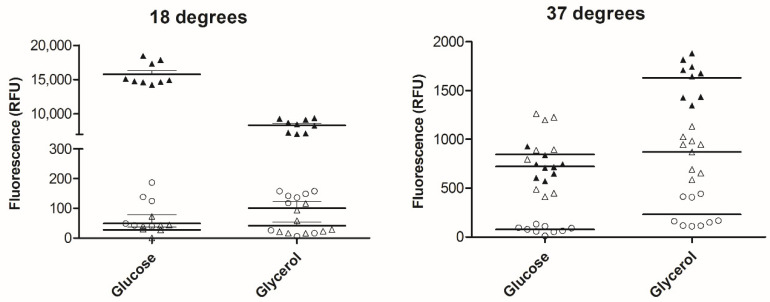
Expression of F9 transcriptional and translational fusions in vitro in defined media. Enterohaemorrhagic *Escherichia coli* (EHEC) Sakai carrying plasmids with the 5′UTR of *fmlA* (pAH010; open triangles) or putative regulator *ECs2114* (pAH011; open circles) cloned upstream of *gfp*+ or 5′UTR with 5 aa CDS of *fmlA* (pACloc8; closed triangles) cloned upstream of *egfp*. Bacteria were grown in rich defined (RD)-MOPS glucose or glycerol at 37 °C or 18 °C shaking for 24 h. Values plotted are corrected for background fluorescence from the promoterless reporter plasmid and normalized for OD_600_. Plasmids are described in Table 1.

**Figure 2 ijms-21-09720-f002:**
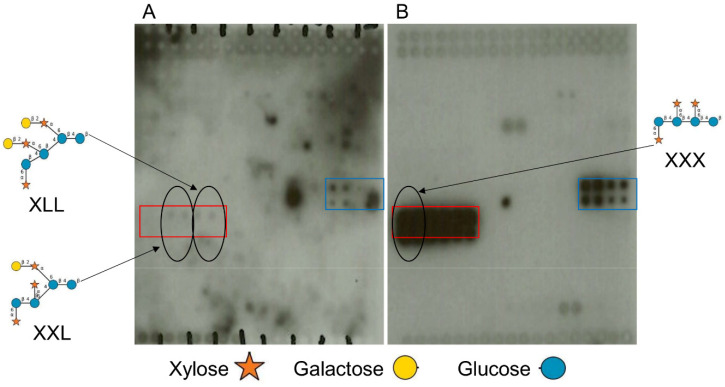
Plant oligosaccharide and polysaccharide interactions with F9. (**A**) Plant glycan arrays (approx. 2.5 cm × 2.5 cm) comprising 81 plant carbohydrates probed with purified F9 detected with a specific anti-F9 antibody on ECL film. Areas with positive interactions are highlighted: red box is BSA-conjugated xyloglucan oligosaccharide, black circles show respective glycan position on the array, and blue box is xyloglucan polysaccharide. The glycans are arranged as described in the Material and Methods; refer to Table 2 for details. (**B**) Glycan arrays probed with anti-xyloglucan antibodies LM15. All glycan symbol structures were made using software DrawGlycan-SNFG [33].

**Figure 3 ijms-21-09720-f003:**
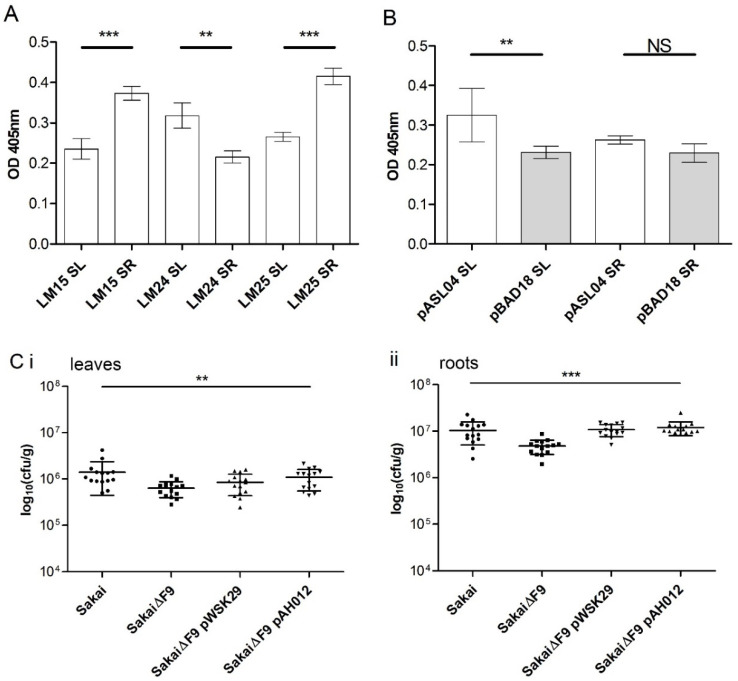
Functional analysis of F9 adherence to spinach extracts and tissues. (**A**) Enzyme linked immuno-sorbent assay (ELISA) of hemicellulose extract from spinach leaves (SLs) and roots (SRs) screening with three hemicellulose probes. (**B**) Bacterial adhesion of *E. coli* expressing F9 (pASL04) or not (pBAD18) on hemicellulose enriched polysaccharides from spinach roots (SRs) and leaves (SLs). Data shown from two independent experiments (mean ± SD; students *t*-test, ** *p* < 0.005, *** *p* ≤ 0.0001, NS = not significant). (**C**) Bacteria were incubated on spinach leaves (**i**) or roots (**ii**) for 2 h in PBS. The number of recovered bacteria was significantly reduced for Sakai lacking F9 fimbriae on both spinach tissue types (one-way analysis of variance (ANOVA); ** *p* < 0.007, *** *p* ≤ 0.0001). Data from three independent experiments with five biological repeats are shown (mean ± SD).

**Figure 4 ijms-21-09720-f004:**
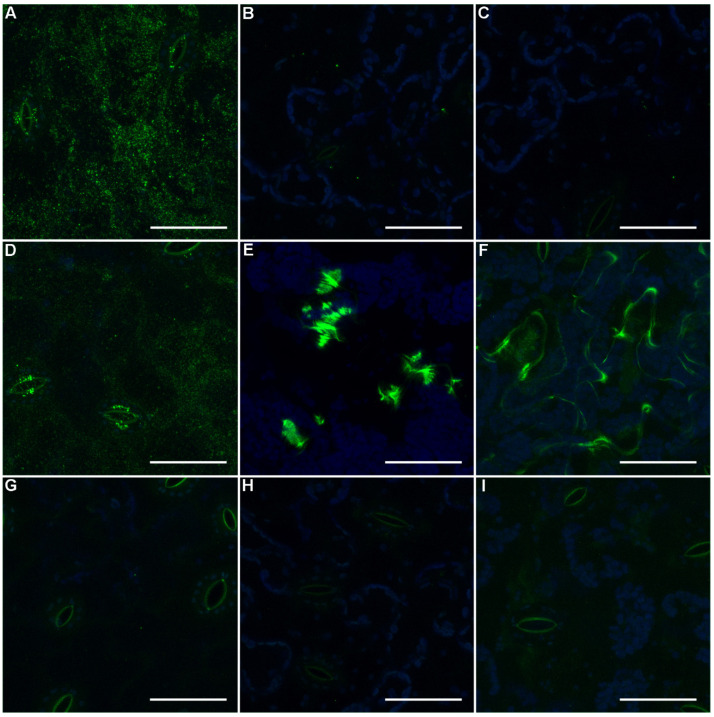
In situ localisation of xyloglucan in spinach. Confocal microscopy images of the abaxial surface of spinach leaves following incubation with LM25 (**A**), LM24 (**B**), or LM15 (**C**); tissue pre-treated with pectate lyase and incubated with LM25 (**D**), LM24 (**E**), and LM15 (**F**); or untreated (**G**), CAPS buffer (**H**), or pectate lyase treated (**I**) tissue incubated without primary antibody followed by Alexa Fluor^®^ 488 secondary antibody (green). All images are maximum intensity projections of z-stacks also showing the autofluorescence from chlorophyll (blue). Scale bars for all images represent 50 µm.

**Figure 5 ijms-21-09720-f005:**
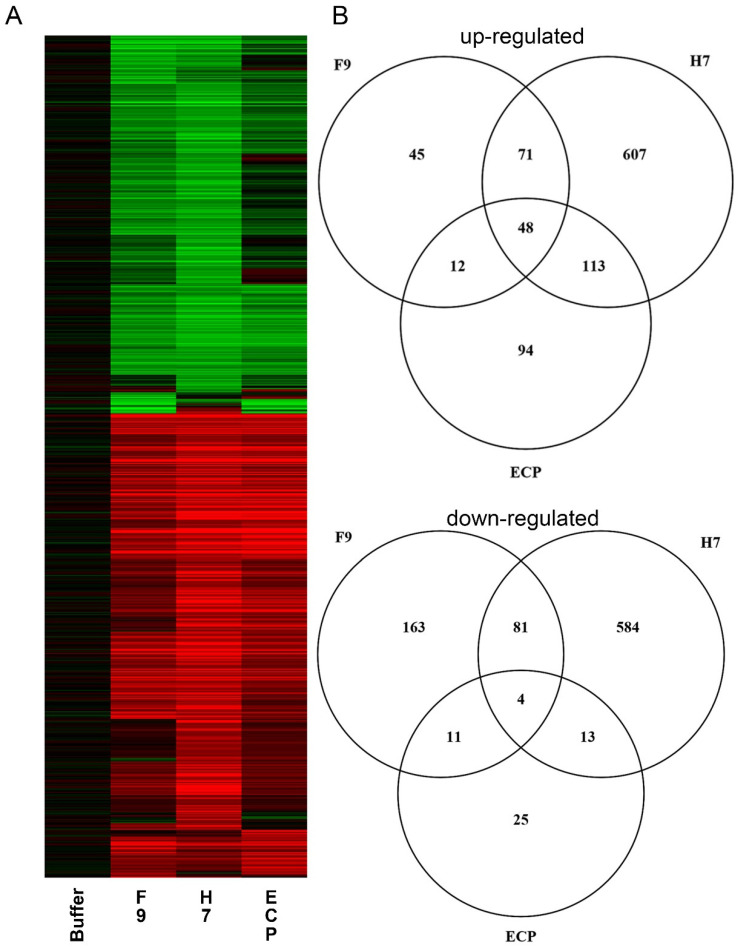
Gene expression overview. (**A**) Heatmap of *Arabidopsis thaliana* transcriptomic responses to F9 fimbriae, H7 flagella, and *E. coli* common pilus (ECP) fimbriae compared with buffer only control. Significant (*p* > 0.05) changes in expression of at least twofold are shown for induced (red) or repressed (green) genes. (**B**) Venn diagrams showing distribution of up-regulated (top) and down-regulated (bottom) genes between the three treatments.

**Figure 6 ijms-21-09720-f006:**
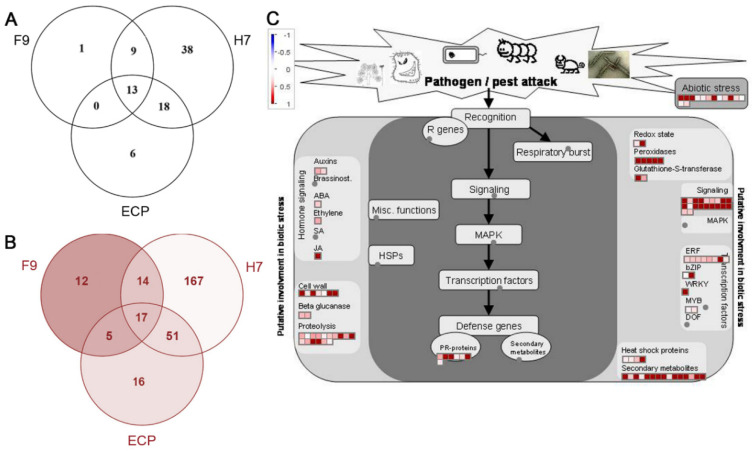
F9 responsive genes common to overall *A. thaliana* responses to (**A**) *E. coli* DH5α [34], (**B**) pathogen-associated molecular patterns (PAMPs) [35], and (**C**) biotic stress mapped by MapMan [36].

**Figure 7 ijms-21-09720-f007:**
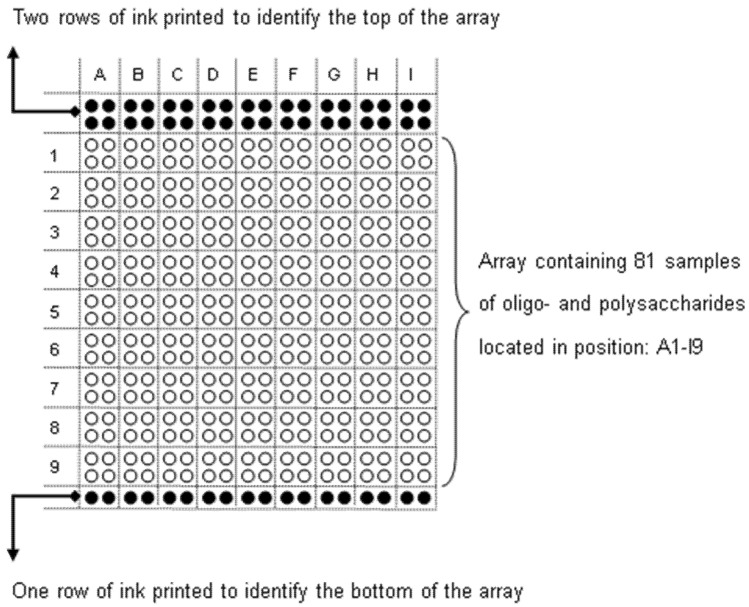
Glycan array layout. Eighty-one samples were spotted in two concentrations (1.0 mg/mL and 0.2 mg/mL) in duplicate. The glycan array is organized in two parts: from A1 to D9, BSA-conjugated oligosaccharides and, from E1 to I9, polysaccharides were spotted. Table 2 lists samples printed in positions A1-I9.

**Table 1 ijms-21-09720-t001:** Plasmids and primers.

**Plasmid**	**Description**	**Reference**
pASL04	pBAD_loc8	[13]
pACloc8	translational reporter of *fmlA*	[13]
pKC026	pAJR145 with P *rpsM* replaced by P *gyrA*. camR	[59]
pAH010	pKC026 with *ECs2113* UTR	this study
pAH011	pKC026 with *ECs2114* UTR	this study
pTOF24	Temperature-sensitive allelic exchange vector	[60]
pTOF61	A plasmid derived from pTOF1 with TcR cloned into SmaI site	[61]
pAH013	pTOF24: PstI-SalI insertion of *ECs2113*. NotI linker + NotI insertion of TcR from pTOF61 CamR TetR Ts SucS	this study
pWSK29	lac operon, low copy number (< 10) ampR	[62]
pAH012	pWSK with *ECs2113* and 500bp flanking sequence	this study
pMAT3	pSE380; ampR; *ecpA–E* (IHE3034)	[63]
pEBW3	pWSK29; ampR; SalI-BamHI insertion of *fliC* (H7, TUV93-0)	[64]
**Primer Name**	**Sequence**	**Construct/qPCR**
2113utr_for	GCTCTAGACTGTCGGTCAGCTTTAAT	pAH010
2113utr_rev	GCTCTAGAAATTTATATTTAAGTAGC	pAH010
2114utr_for	GCTCTAGAGTATTAAGTAAGAGTAAGATATGGG	pAH011
2114utr_rev	GCTCTAGATTTGCCAGCTTGAGTCTT	pAH011
2113.SalI.5F	TTATCGTCGACCCGACGGGAATATTTGCGGATG	pAH013
2113.NotI.5R	CCGTTCCAAGCGGCCGCAAGAGCGAATTCTCCAAAATTTATATTTAAG	pAH013
2113.NotI.3F	CGCTCTTGCGGCCGCTTGGAACGGTTCAATATTAAGGTGGCTATACG	pAH013
2113.PstI.3R	GGATCCTGCAGCGTGACAGGCAATAAAACCTG	pAH013
2113.5.F	CTGTCGGTTACGAAAGCTTATC	pAH012
2113.3.R	GGCAGGATCATTATTCGTGACAGG	pAH012
Actin1.F	TCAGGTAGAAGAAAATGGCTGA	qPCR
Actin1.R	TAGGTGCATCATCCCCAGC	qPCR
MAPKKK15.F	CAGACTGGGAAACGAACGGT	qPCR
MAPKKK15.R	TCCTCCTCCAATGACGTTGC	qPCR
AT4G12500.F	ATCAGCTCAACCATGCTGCT	qPCR
AT4G12500.R	CCTAAGGGCAGTGCAGAGAC	qPCR
AT1G22900.F	CGCATCGACGGACATGAAAA	qPCR
AT1G22900.R	GGTGCTGCCGTTAAACTCTC	qPCR
AT5G44910.F	CAATAGTAGAGCGGACCGGC	qPCR
AT5G44910.R	CTTAACCGCCGTGACGATTG	qPCR
AT1G10360.F	TCTGTGATGATCCCGCTGTG	qPCR
AT1G10360.R	CCAGTTTATGCTTGCCGCTT	qPCR
AT2G17740.F	AGCTGCAATGTTAAAGGCACA	qPCR
AT2G17740.R	AGCTCTTTGCATCTCCAAGTGA	qPCR
WRKY29.F	GAAACGAGTACGCACCAAGC	qPCR
WRKY29.R	CTCCCGGACATCAAATCCGA	qPCR
AT1G33760.F	TCCATTAGACTCGCCGAGGA	qPCR
AT1G33760.R	CATCCCTATCGTGCTGACCA	qPCR
AT1G24020.F	ATGGAGAAGGATCTCCACTGGT	qPCR
AT1G24020.R	CGCCAATGATGCTGTACGAC	qPCR
AT3G61930.F	AATCGGACGACGGTGGTAAG	qPCR
AT3G61930.R	TGAGATCGAACATCGCCACC	qPCR

**Table 2 ijms-21-09720-t002:** List of plant oligosaccharides and polysaccharides. The letter and number code relates to the position printed on the glycan array shown in Figure 7.

Sample List			
A1	α-(1→4)-D-heptagalacturonate (DE 0%)	F5	Arabinoxylan (wheat)
B1	α-(1→4)-D-pentagalacturonate (DE 0%)	G5	4-Methoxy-glucoronoarabinoxylan (birch)
C1	β-D-galactose	H5	Xyloglucan #1 (tamarind)
D1	β-(1→4)-D-galactotetraose	I5	Xyloglucan #2 (pea)
E1	Pectin (lime, DE 11%)	A6	Xyloglucan XXX(G)-
F1	Pectin (lime, DE 43%)	B6	Xyloglucan XXL(G)-
G1	Pectin (lime, DE 0%)	C6	Xyloglucan XLL(G)-
H1	Pectin (lime, DE 16%)	D6	6-α-D-galactosyl-β-(1→4)-D-mannobiose
I1	Pectin (sugar beet)	E6	Carboxymethyl-cellulose
A2	α-L-arabinose	F6	Hydroxymethyl-cellulose
B2	α-(1→5)-L-arabinobiose	G6	Hydroxyethyl-cellulose
C2	α-(1→5)-L-arabinotriose	H6	(1→6):(1→3)-β-D-glucan (*Laminaria digitata*)
D2	α-(1→5)-L-arabinotetraose	I6	(1→3)-β-D-glucan (Poria cocos)
E2	Pectic galactan #1 (lupin)	A7	(1→3),(1→4)-β-D-glucotriose [G4G3G(G)-]
F2	Pectic galactan #2 (potato)	B7	(1→3),(1→4)-β-D-glucotetraose [G3G4G4G(G)-]
G2	Pectic galactan #3 (lupin)	C7	(1→3),(1→4)-β-D-glucopentaose [G3G4G3G4G(G)-]
H2	Pectic galactan #4 (tomato)	D7	6^3^,6^4^-*di*galactosyl-β-(1→4)-D-mannotetraose
I2	Pectin (RGII enriched, red wine)	E7	β-glucan #1 (lichenan, icelandic moss)
A3	α-(1→5)-L-arabinopentaose	F7	β-glucan #2 (barley)
B3	α-(1→5)-L-arabinohexaose	G7	β-glucan #3 (oat)
C3	α-(1→5)-L-arabinoheptaose	H7	β-glucan #4 (yeast)
D3	β-(1→4)-D-glucopentaose	I7	(1-6);(1-4)-α-D-glucan (*Aureobasidium pullulans*)
E3	RGI #1 (soy bean)	A8	α-(1→4)-D-glucobiose
F3	RGI #2 (carrot)	B8	α-(1→4)-D-glucopentaose
G3	RGI #3 (sugar beet)	C8	α -(1→6)-D-glycosyl-α-(1→4)-D-maltotriose
H3	RGI #4 (arabidopsis)	D8	α-(1→6)-D-glycosyl-α-(1→4)-D-maltosyl-maltose
I3	Seed mucilage (arabidopsis)	E8	Glucomannan (konjac)
A4	β-D-mannose	F8	Galactomannan (carob)
B4	β-(1→4)-D-mannotriose	G8	Gum (guar)
C4	β-(1→4)-D-mannotetraose	H8	Gum (arabic)
D4	β-(1→4)-D-mannopentaose	I8	Gum (locust bean)
E4	Arabinan #1 (sugar beet)	A9	β-D-glucose
F4	Arabinan #2 (sugar beet)	B9	β-(1→3)-D-glucotriose
G4	Arabinan #3 (sugar beet)	C9	β-(1→3)-D-glucotetraose
H4	RGI #5 (potato)	D9	β-(1→3)-D-glucopentaose
I4	Xylogalacturonan (apple)	E9	Gum (xanthan)
A5	β-D-xylose	F9	Carrageenan #1 (red seaweed)
B5	β-(1→4)-D-xylotriose	G9	Carrageenan #2 (red seaweed)
C5	β-(1→4)-D-xylotetraose	H9	Alginate (brown seaweed)
D5	β-(1→4)-D-xylopentaose	I9	RGI #6 (arabinan enriched, sugar beet)
E5	Xylan (birch)	DE; degree of methyl esterification	

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
