# Peer review of "Escherichia coli O157:H7 F9 Fimbriae Recognize Plant Xyloglucan and Elicit a Response in Arabidopsis thaliana"

_ijms, 2020, doi:10.3390/ijms21249720_

Round 1

Reviewer 1 Report

see below:

I read the paper and came to the conclusion that is is too biological for me. The topic appeals to me as we did a lot of work on bacterial adhesion and E. coli is an important target in this respect.  Also I think that is a good thing the authors did a carbohydrate microarray.  However, it is not clear at all which sugars were on the array and which were recognized.  The nomenclature they use does not work for me.  I need molecular structure or at least schematic structures for glycans. I am not sure if this is my problem as an organic chemist.  I hope these comments clarify my point and I hope you can put them in perspective.

Author Response

We thank the reviewer for their honest comments.  The nomenclature in the manuscript has been adopted in other publications describing the use of these carbohydrate arrays (Moller et al (2008) Glycoconj J 25(1): 37–48. doi: 10.1007/s10719-007-9059-7).  We have added a structural schematic to Figure 2 in order to improve the interpretation of the results from a non-biological perspective.

Reviewer 2 Report

Please correct the following errors:

Line 214: Figure 6A, not Figure 5A

Line 218: Figure 6B, not Figure 5B

Line 224: Figur 6C, not Figure 5C

Author Response

We would like to thank the reviewer for taking the time to evaluate our manuscript and bring the aforementioned typing errors to our attention.

Reviewer 3 Report

IJMS-1008264

Holmes et al

The authors describe F9 fimbriae by EHEC as attachment tool to plant cell wall, in particular galactosylated chains of xyloglucan of spinach. This subsequently leads to the transcriptional regulation of many Arabidopsis thaliana genes. The research is significant and relevant because of plant/produce caused outbreaks of EHEC. I have a question on the experimental design on the second part (transcriptomics) of the study.  The manuscript is written in a somewhat confusing style as introduction, methods, and results are mixed up and not separated properly. Specific examples are given under minor comments.

Major comments:

  • Why was the first experiment done with spinach and the second with A. thaliana? Would not it make more sense to do the entire study with the same plant? I understand A. thaliana is a model and maybe the microarray was available. But you can do mRNA sequencing with any organism these days.

Minor comments:

  • Lines 83 to 97: There is introduction at the beginning of the results sections. Get that into the introduction. Though there is a Fig. S1 here. Everything that is referenced needs to be in the introduction. Figures and supplemental figures are results.
  • Lines 99 to 112: there is methods in the results. Get the fusion construction into the methods.

Author Response

·         Why was the first experiment done with spinach and the second with A. thaliana? Would not it make more sense to do the entire study with the same plant? I understand A. thaliana is a model and maybe the microarray was available. But you can do mRNA sequencing with any organism these days.

While this is a valid point, we took this route for two reasons: firstly as mentioned, A. thaliana is an established model, with a huge set of experimental resources available, including the DNA microarray used here (we have published before on the validity of microarray compared to RNAseq for transcriptomes).This dataset allowed direct comparisons of the responses to published datasets, whereas at the time of study, the draft genome of spinach was not available. Secondly, we used spinach for the adherence assays to demonstrate F9 provides functional binding, since it is a known vehicle of EHEC transmission and therefore relevant horticultural crop plant species.

Minor comments:

·         Lines 83 to 97: There is introduction at the beginning of the results sections. Get that into the introduction. Though there is a Fig. S1 here. Everything that is referenced needs to be in the introduction. Figures and supplemental figures are results.

·         Lines 99 to 112: there is methods in the results. Get the fusion construction into the methods.

Thank you for the comments to improve the manuscript.  We have moved the sections to Introduction (Lines 52-59) and Methods (Lines 321-325) as recommended.

Reviewer 4 Report

In this manuscript entitled “Escherichia coli O157: H7 F9 fimbriae recognize plant xyloglucan and elicit a response in Arabidopsis thaliana.”, Holmes et al. initially investigated the glycan-binding specificity of F9 fimbriae using glycan arrays. Fimbriae are extracellular structures of E. coli that are involved in cellular attachment and surface colonization. The authors showed that F9 fimbriae bind plant cell wall hemicellulose, specifically galactosylated side chains of xyloglucans. Next, the authors investigated whether F9 fimbriae induces a transcriptional response in Arabidopsis thaliana as a well-established model for the analysis of plant defense response, compared to H7 flagella and E.coli common pilus (ECP) fimbriae. They found that F9 induces the differential expression of 435 genes including some genes which are involved in plant defense response. Based on these results, the authors concluded that “The expression of F9 at environmentally relevant temperatures and its recognition on plant xyloglucan adds to the suite of adhesions enterohemorrhagic E. coli has available to exploit the plant niche”.

Although this study contains interesting content, there are areas for improvement in the methods of data analysis and the structure of the article. Therefore, I cannot recommend that it be accepted in its current form. Please respond appropriately to the following comments.

(1) In Figure 3A, the authors compared the reactivity of each glycan-specific antibody in spinach leaves (SL) and roots (SR). They should indicate how they normalized the use of each fraction in the ELISA assays to ensure a fair comparison.

(2) In Figure 3B, the authors describe “Data shown from two independent experiments (Mean+/-SD).” This is strange to me. How could the authors calculate the SD value only from two independent experiments?

(3) It is difficult to tell which sample is being compared to which test results in Figure 3C. Currently, it appears that only the results of the comparison of the two groups are displayed. However, in the main text, the authors describe the results of the comparison between more groups. What is shown in the main text should also be shown in Figure 3C.

(4) In Figure 4, the authors should add a scale bar to each panel.

(5) Line 201 about Figure 5, the sentence “Only 1.2% (23) of differentially expressed genes were shared by F9 and ECP” does not seem to be accurate because it does not add the property that there was no change in gene expression in H7. The authors should improve this sentence.

(6) The first paragraph of Section 2.4.2, i.e., lines 210-222, lists previously published data and not the results obtained in this study. Therefore, this section should be moved to the discussion section.

Author Response

(1) In Figure 3A, the authors compared the reactivity of each glycan-specific antibody in spinach leaves (SL) and roots (SR). They should indicate how they normalized the use of each fraction in the ELISA assays to ensure a fair comparison.

Line 373 Polysaccharide content was quantified by a phenol–sulfuric acid method in microplate format (Masuko T et al., 2005) after acid hydrolysis (2 M trifluoroacetic acid for 1 h at 120 °C).

Line 377 NUNC Maxisorp 96 well plates were coated with 5μg polysaccharides (in Tris buffered saline (TBS)) overnight at 4°C then incubated with ~1x106 cfu bacteria or 1/20 dilution monoclonal antibody.

(2) In Figure 3B, the authors describe “Data shown from two independent experiments (Mean+/-SD).” This is strange to me. How could the authors calculate the SD value only from two independent experiments?

Each experiment comprised of at least three technical replicates and were repeated on two separate occasions.

(3) It is difficult to tell which sample is being compared to which test results in Figure 3C. Currently, it appears that only the results of the comparison of the two groups are displayed. However, in the main text, the authors describe the results of the comparison between more groups. What is shown in the main text should also be shown in Figure 3C.

To clarify the results shown, the terms leaves and roots have been added to Figure 3 to make it clearer to the reader.  We show that the fmlA mutant is significantly reduced in its ability to adhere to both spinach leaf and root tissues.  There are some differences in the results obtained with the complemented mutant between the tissue types which we describe in the main text based upon the post hoc statistical tests.

(4) In Figure 4, the authors should add a scale bar to each panel.

 A scale bar has been added to each panel.

(5) Line 201 about Figure 5, the sentence “Only 1.2% (23) of differentially expressed genes were shared by F9 and ECP” does not seem to be accurate because it does not add the property that there was no change in gene expression in H7. The authors should improve this sentence.

The text has been updated as follows: Only 1.2% (23) of differentially expressed, H7-independent, genes were shared by F9 and ECP; these genes may be described as fimbrial-responsive

(6) The first paragraph of Section 2.4.2, i.e., lines 210-222, lists previously published data and not the results obtained in this study. Therefore, this section should be moved to the discussion section.

This paragraph introduces the published dataset that we compared our data to and references the results from this comparison in Table S2.  We argue that this paragraph remains within the results section.

We would like to thank the reviewer for their careful consideration of our manuscript and constructive comments.

Round 2

Reviewer 3 Report

I am okay with the author's explanation on the two different types of plants used. They did move the introductory portions that were in the results and methods up front.